# Scalp Melanoma: A High-Risk Subset of Cutaneous Head and Neck Melanomas with Distinctive Clinicopathological Features

**DOI:** 10.3390/jcm12247643

**Published:** 2023-12-13

**Authors:** Rodolfo David Palacios-Diaz, Blanca de Unamuno-Bustos, Mónica Pozuelo-Ruiz, Enrico Giorgio Morales-Tedone, Rosa Ballester-Sánchez, Rafael Botella-Estrada

**Affiliations:** 1Department of Dermatology, Hospital Universitari i Politècnic La Fe, 46026 Valencia, Spain; rodolfo.palaciosd@gmail.com (R.D.P.-D.); m.pozueloruiz@hotmail.com (M.P.-R.); rbotellaes@gmail.com (R.B.-E.); 2Department of Dermatology, Instituto de Investigación Sanitaria La Fe, 46026 Valencia, Spain; 3Department of Dermatology, Hospital Clínico Universitario de Valencia, 46010 Valencia, Spain; enricogmt@gmail.com (E.G.M.-T.); roseta999@hotmail.com (R.B.-S.); 4Department of Medicine, Universitat de València, 46010 Valencia, Spain

**Keywords:** skin cancer, melanoma, prognosis, sentinel lymph node biopsy

## Abstract

Scalp melanomas (SM) have been previously associated with poor overall and melanoma-specific survival rates. The aim of this study was to describe and compare the clinicopathological characteristics and survival outcomes of SM and non-scalp cutaneous head and neck melanoma (CHNM). An observational multi-center retrospective study was designed based on patients with CHNM followed in two tertiary care hospitals. A hundred and fifty-two patients had CHNM, of which 35 (23%) had SM. In comparison with non-scalp CHNM, SM were more frequently superficial spreading and nodular subtypes, had a thicker Breslow index median (2.1 mm vs. 0.85 mm), and a higher tumor mitotic rate (3 vs. 1 mitosis/mm^2^) (*p* < 0.05). SM had a higher risk of recurrence and a higher risk of melanoma-specific death (*p* < 0.05). In the multivariate analysis, scalp location was the only prognostic factor for recurrence, and tumor mitotic rate was the only prognostic factor for melanoma-specific survival. We encourage routinely examining the scalp in all patients, especially those with chronic sun damage.

## 1. Introduction

Previous studies have demonstrated an association between the anatomic location of primary cutaneous melanoma and survival [1]. Classically, stage II melanomas located in the back, arm, neck, and scalp regions appeared to have a poorer prognosis than those in different locations [2,3]. More recently, the axial site (trunk, head, and neck) has shown a worse melanoma prognosis over the extremities [4].

Among locations with poorer outcomes, cutaneous head and neck melanomas (CHNM) are of special interest. Head and neck denote an anatomic area of 9% of the whole-body surface; however, previous reports have found that between 11 and 26.7% of all cutaneous melanomas are located in this region [5,6]. Moreover, the head and neck are chronically sun-exposed areas, and CHNM occurs in older people and has a worse prognosis than melanomas at other sites [6]. In addition, even within head and neck locations, different clinical and histopathologic characteristics and survival have been found [6,7].

Scalp melanomas (SM) represent between 24.9 and 35% of CHNM [5,7,8]. They have been associated with reduced melanoma-specific and overall survival compared to non-scalp CHNM, as well as melanomas located in the trunk and extremities [9]. Thus, melanomas arising in the scalp might be a high-risk subset of CHNM. The present study aimed to describe the clinical and histopathologic characteristics of SM and to compare them with non-scalp CHNM. Furthermore, we analyzed recurrence and survival between these two groups.

## 2. Materials and Methods

We designed an observational, multicenter, and retrospective study. Data on patients were collected from the databases of the Dermatology Departments of two tertiary-level referral hospitals located in Valencia, Spain: Hospital Universitari i Politècnic La Fe and Hospital Clínico Universitario de Valencia. Information was compiled from January 2014 to December 2022. All patients with primary CHNM, either in situ or invasive, were eligible for analysis. Patients with mucosal melanoma or primary unknown cutaneous melanoma were excluded from the analysis.

Epidemiological (sex, age, Fitzpatrick Skin Phototype, history of severe sunburns, and chronic sun exposure) and clinical (presence of freckling, lentigines, actinic keratosis, common nevi, family history of melanoma, history of multiple primary melanomas, melanoma location, and ancillary exams, including lympho-gammagraphy and sentinel lymph node biopsy) data were systematically obtained during the medical visit and were documented in the patients’ medical records. Histopathological data (histologic subtype, Breslow thickness, ulceration, lymphocyte infiltration, tumor mitotic rate, regression, underlying histologic lesion, and solar elastosis) was obtained from the standardized pathology record. The anatomical site was classified into scalp and non-scalp. The scalp limits were established as the forehead anteriorly (4 cm superior to the supraorbital ridge), the superior nuchal line posteriorly, and the zygomatic arch and external acoustic meatus laterally [10]. If a melanoma was in the hairline and at least 50% of its surface was in the scalp, it was considered an SM [11]. Given the uncertain biological behavior and the absence of general consensus, intermediate malignant tumors (such as SAMPUS and MELTUMP) were not considered in the dataset.

Quantitative variables were expressed as mean and standard deviation, or median and 25–75th percentiles, depending on the normality of the distribution of the variable. Qualitative variables were characterized by absolute and relative frequencies. The comparison between the qualitative variables was accomplished with the chi-square test, and the quantitative variables were compared with the Mann–Whitney U test. Analysis of recurrence-free survival and melanoma-specific survival was performed with Kaplan–Meier survival estimates and Cox proportional hazards models. For the interpretation of the results, *p*-values less than 0.05 were considered significant. Statistical analysis was carried out using Microsoft Excel and Stata, version 17.0. The study was conducted according to the guidelines of the Declaration of Helsinki and approved by the Institutional Review Board of Instituto de Investigación Sanitaria La Fe (IISLaFe).

## 3. Results

### 3.1. Clinical and Epidemiological Characteristics

Information was available from 693 patients diagnosed with melanoma during the period of data collection. Of these, 152 (21.9%) patients had a CHNM. Among the patients with CHNM, 35 melanomas (23%) were located on the scalp. Table 1 describes the epidemiological and clinical characteristics of the patients with SM and non-scalp CHNM. Overall, most patients with CHNM (94/152; 61.8%) were men. Additionally, the patients with SM were less frequently women (22.9% vs. 42.7, *p* = 0.034). The patients with SM had a median age of 71 years, and they had signs of chronic actinic damage, such as solar lentigines (27/35; 77.1%) and actinic keratosis (17/35; 48.6%).

### 3.2. Histopathological Characteristics

Although lentigo maligna melanoma (LMM) was the most frequent histological subtype in CHNM (65.1%), superficial spreading melanoma (SSM) and nodular melanoma (NM) were statistically more frequent in patients with SM (Table 1). Moreover, in comparison with non-scalp CHNM, SM were more frequently invasive (77.1 vs. 47%, *p* = 0.002) and had a higher Breslow index median (2.1 mm vs. 0.85 mm, *p* = 0.002). When further classified, thin melanomas (≤1 mm) were more frequent in non-scalp CHNM (56.3% vs. 18.5%, *p* = 0.001). When looking for further differences in Breslow thickness, we sorted the patients with SM by age lower and higher than 71 years, based on the median age. Although there was a tendency towards a lower median in patients with an age lower than 71 years, we did not find statistically significant differences (1.7 mm vs. 2.5 mm, *p* = 0.678).

SM had a higher tumor mitotic rate than non-scalp CHNM (3 vs. 1 mitosis/mm^2^, *p* = 0.012). Ulceration was found more often in SM (22.9% vs. 10.3%), but it did not reach statistical significance (*p* = 0.05). Neurotropism was found in only two non-scalp CHNMs, and one SM had satellitosis. No statistically significant differences were found regarding other histopathological characteristics.

### 3.3. Survival Analysis and Prognostic Factors

The median follow-up was 41.6 months (25–75th percentile: 18–66.7 months). There was no statistically significant difference between the time of follow-up for SM and non-scalp CHNM (44 vs. 41.6 months; *p* = 0.876).

Lympho-gammagraphy and sentinel lymph node biopsy were performed in 25 patients. One patient with SM had a positive sentinel lymph node biopsy (1/9, 11.1%). Moreover, the patients with SM had a significantly higher rate of non-identified lymph nodes due to abnormal tracer migration (55.6% vs. 18.8%, *p* = 0.043) (Table 1). Eleven patients (7.2%) of the whole cohort of CHNM recurred. Recurrence was significantly higher in patients with SM (22.9% vs. 2.6%, *p* < 0.001). The patients with SM had a first recurrence of melanoma at the local (42.9%, 3/7) and distant sites (57.1%, 4/7). There was no information regarding the recurrence of one patient with SM. The patients with non-scalp CHNM had a first recurrence of melanoma at local (33.3%, 1/3), regional (33.3%, 1/3), and distant sites (33.3%, 1/3). 

Kaplan–Meier curves for disease-free survival showed a significantly lower survival for SM (log-rank test *p* < 0.001) (Figure 1). In univariable analysis, scalp location, histologic subtype (SSM), ulceration, peritumoral lymphocyte infiltration, higher tumor mitotic rate, and sentinel lymph node biopsy were associated with a higher risk of recurrence (*p* < 0.05) (Table 2). After controlling for location, histologic subtype (SSM), ulceration, peritumoral lymphocyte infiltration, and higher tumor mitotic rate in a multivariate Cox model, only scalp location remained a significant risk factor for melanoma recurrence (hazard ratio (HR) 12.857; *p* = 0.002). We did not consider sentinel lymph node biopsy because, given the small proportion of patients that had this procedure done, the model was not representative of the whole population.

The patients with SM had significantly lower melanoma-specific survival in Kaplan–Meier curves (log-rank test; *p* = 0.027) (Figure 2). In univariable analysis, a higher tumor mitotic rate and histologic subtype (SSM) were associated with a higher risk of melanoma-specific death (*p* < 0.05) (Table 2). After controlling for these variables in a multivariate Cox model, only the higher tumor mitotic rate remained a significant risk factor for melanoma-specific survival (HR 1.505; *p* = 0.013).

## 4. Discussion

Head and neck location is of special interest when considering anatomic site as a prognostic factor in cutaneous melanomas [6]. In this study, we described the clinical and histopathological characteristics of a cohort of patients with CHNM in a Spanish Mediterranean population. In our cohort, 21.9% of the patients developed cutaneous melanoma in either the head or neck. This frequency is comparable to previous reports [1,5,8,9]. Considering its surface area, a higher density ratio of cutaneous melanoma has been found in the head and neck than in other sites [5,6]. Moreover, head and neck locations have been found to be frequent sites for second primary melanomas, and a high location concordance for first and second melanomas has been reported [12,13]. This highlights the importance of a careful skin examination of this area.

In line with previous studies, our results showed that patients with either scalp or non-scalp CHNM were mostly elderly men [14,15,16]. Our patients also had clinical signs of long-term sun exposure, such as lentigines and actinic keratosis. This has been scarcely reported before [1,8]. Furthermore, there is a trend towards an increase in the incidence of CHNM as well as thick melanomas in older cohorts with higher cumulative sun exposure [17,18]. Given the current trends and its typical consideration as a chronic sun-exposed area, the head and neck sites are of sensible importance.

When compared with non-scalp CHNM, SM were more frequently SSM and NM and less frequently LMM (*p* < 0.05). This was further supported by an increased frequency of invasiveness and a thicker Breslow index. The data regarding histological subtypes in SM are conflicting. Most authors have reported high rates of SSM and/or NM in the scalp [15,19,20]. Conversely, others have reported higher rates of LMM [8,16]. Although the desmoplastic histological subtype has been shown to be associated with SM, we only found a desmoplastic melanoma in the non-scalp CHNM cohort (1/117) [8,15]. 

Classically, SSM has been considered within the pattern of low cumulative sun-damaged melanomas [21]. On the other hand, LMM results from chronic and cumulative sun exposure, overcoming the mechanism of photoadaptation [6,21]. This explains the occurrence of LMM in older cohorts of patients and its typical location in sun-exposed areas such as the face. Given that the scalp is covered by hair, at least earlier in life, UV exposure in the first decades of life might be prevented, and classical mechanisms of photoadaptation might be less developed in the scalp. This may partially explain the more frequent histological subtypes of SSM and NM affecting the scalp.

In our study, the recurrence rate for SM was 22.9%. The patients with SM had a significantly higher hazard of recurrence. This was not explained despite adjustments for histologic subtype, ulceration, peritumoral lymphocyte infiltration, and tumor mitotic rate, which initially showed a significant hazard ratio in the univariable analysis. Previous studies support the finding that the scalp is a risk factor for recurrence [9,19,22]. Furthermore, the rate of recurrence for SM is higher than that for non-scalp head and neck locations, trunks, and extremities [23]. In our cohort, SM had a similar rate of first-time disease recurrence in local and distant sites. Previous studies have reported variable rates of local, regional, and distant recurrences [10,16]. The complex lymphatic drainage of this area may lead to an unpredictable risk of distant or local recurrence [9,10]. Moreover, resection above the galea has been found to be associated with worse disease-free survival [10].

As previously reported by other authors, we found poorer melanoma-specific survival in SM [1,7,9,24,25]. After adjusting for significant variables, tumor mitotic rate remained the only prognostic factor for melanoma-specific survival in the multivariate model. Tumor mitotic rate has been found to be an independent predictor of overall and disease-specific survival, even after stratifying by clinical stage [16,26]. Thus, the higher mortality rate in SM might be related to the more frequent presence of mitosis in our cohort. Other established adverse prognostic factors, such as deep Breslow depth, ulceration, and nodular subtype, were more frequent in SM, but they were not independent risk factors.

Lympho-gammagraphy and sentinel lymph node biopsy were performed in 30.5% of the patients with invasive CHNM and in 42.6% of the patients with CHNM and Breslow thickness higher than 0.8 mm [27]. Strikingly, after lympho-gammagraphy, only in 68% of the patients was a lymph node identified and successfully excised. This contrasts with the higher rates of sentinel node identification in the head and neck in previous reports [28]. Consistent with previous authors, we found a low rate of positive sentinel lymph nodes for head and neck melanomas [29]. On the other hand, false negative results have been found to be higher in head and neck melanomas compared with other body regions [28,29].

Sentinel lymph node status has been found to be a strong predictor of overall survival and melanoma-related death in SM [25,30]. In our cohort, sentinel lymph node status was found to be a prognostic factor for disease recurrence in univariable analysis. Nonetheless, we did not consider it in the multivariate analysis given the low rate of patients for whom this procedure was performed. Moreover, we found that the patients with SM had a significantly higher rate of non-identified sentinel lymph nodes than patients with non-scalp CHNM, probably due to an unpredictable drainage pattern. It has been found that a posterior SM has frequent scalp and posterior neck sentinel lymph nodes, and a frontal SM frequently has parotid sentinel lymph nodes [31]. Furthermore, certain challenges have been identified when performing sentinel lymph nodes, such as a high discordance rate of positive nodes and hottest nodes, multiple sentinel lymph nodes per patient, and the possibility of contralateral scalp nodes [31]. The surgeons and radiologists should be aware of these to properly identify sentinel lymph nodes for study.

Our study has certain limitations. Our relatively small sample size for recurrence and melanoma-specific death might limit the significance of the multivariate analysis. Thus, we advise careful consideration of the results for the possibility of bias. Given the retrospective design, data were not fully available for some patients. Furthermore, despite a common adherence to current practice guidelines, follow-up, treatment, and complementary studies might not be completely similar in both institutions. In addition, although most patients had complete data regarding histopathologic characteristics, the study of genetic mutations was lacking for most of the patients. Future studies should include genetic studies to elucidate possible mechanisms related to the worse prognosis of SM.

## 5. Conclusions

In summary, we reported a large cohort of patients with CHNM. The patients with SM had significantly higher Breslow thickness, a higher mitotic rate, lower disease-free survival, and higher melanoma-specific mortality. In the multivariate analysis, scalp location remained the only prognostic factor for recurrence, and a higher tumor mitotic rate was the only prognostic factor for melanoma-specific survival. Given the results, we encourage routinely examining the scalp in all patients, especially in old men with signs of chronic sun exposure. In addition, in order to improve screening in the general population, we recommend raising awareness about SM among professionals involved in caring for this area, such as hairdressers.

## Figures and Tables

**Figure 1 jcm-12-07643-f001:**
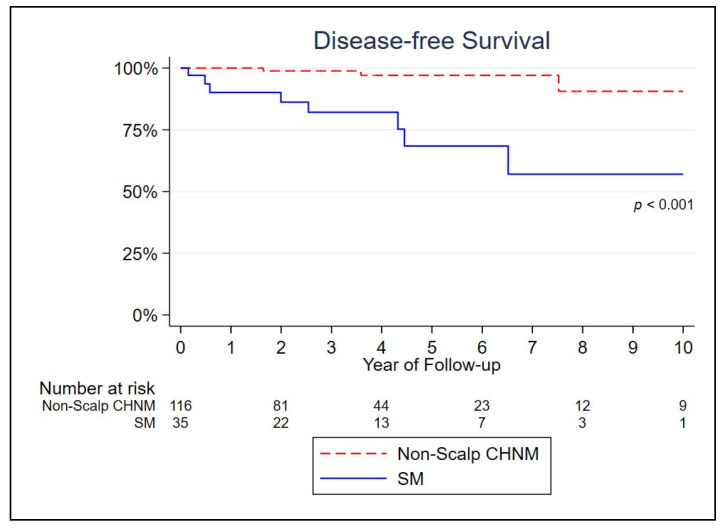
Kaplan–Meier curve of disease-free survival by anatomic site.

**Figure 2 jcm-12-07643-f002:**
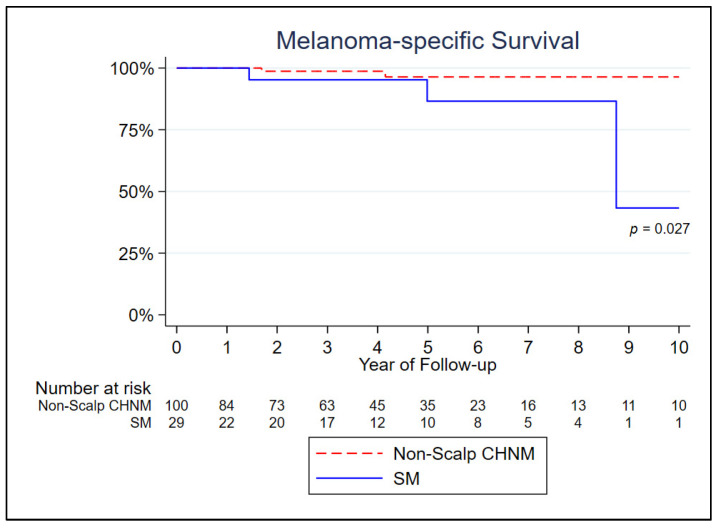
Kaplan–Meier curve of melanoma-specific survival by anatomic site.

**Table 1 jcm-12-07643-t001:** Epidemiological, clinical, and histopathological features of patients with scalp and non-scalp cutaneous head and neck melanoma.

	CHNM (n = 152)						
	Scalp Melanoma (n = 35)		Non-Scalp CHNM (n = 117)		Total		
	n	%	n	%	n	%	*p*
Epidemiological and clinical features							
Sex							
Man	27	77.14	67	57.26	94	61.84	**0.034** ^a^
Woman	8	22.86	50	42.74	58	38.16	**0.034** ^a^
Age (median/25–75th percentile)	71	(59–81)	74	(63–81)	73.5	(63–81)	0.656 ^b^
Fitzpatrick Skin Phototype							
I–III	30	85.71	105	89.75	135	88.81	0.262 ^a^
IV–V	5	14.29	12	10.25	17	11.19	
Severe sunburns	11	31.43	55	47.01	66	43.42	0.103 ^a^
Chronic sun exposure	11	31.43	48	41.03	59	38.82	0.307 ^a^
Freckling	1	2.86	10	8.55	11	7.24	0.254 ^a^
Lentigines	27	77.14	85	72.65	112	73.68	0.596 ^a^
Actinic keratosis	17	48.57	49	41.88	66	43.42	0.483 ^a^
History of non-melanoma skin cancer	8	22.86	25	21.37	33	21.71	0.851 ^a^
Nevi count							
<50	34	97.14	110	94.02	144	94.74	0.551 ^a^
>50	1	2.86	7	5.98	8	5.26	
Family history of melanoma	1	2.86	11	9.4	12	7.89	0.208 ^a^
Multiple primary melanomas	5	14.29	14	11.97	19	12.5	0.716 ^a^
Lympho-gammagraphy and sentinel lymph node biopsy							
Performed	9	25.71	16	13.68	25	16.45	0.092 ^a^
Negative	3	33.33	13	81.25	16	64	**0.043** ^a^
Positive	1	11.11	0	0	1	4	
Not identified	5	55.56	3	18.75	8	32	
Recurrence	8	22.86	3	2.56	11	7.24	<**0.001** ^a^
Death	9	25.71	18	15.38	27	17.76	0.161 ^a^
Melanoma-related	3	8.57	2	1.71	5	3.29	**0.046** ^a^
Non-melanoma-related	6	17.14	16	13.68	22	14.47	0.609 ^a^
**Histopathological features**							
Histologic subtype							
LMM	12	34.29	87	74.36	99	65.13	<**0.001** ^a^
SSM	17	48.57	21	17.95	38	25	<**0.001** ^a^
NM	6	17.14	6	5.13	12	7.89	**0.021** ^a^
Other	0	0	3	2.56	3	1.97	0.339 ^a^
Melanoma in situ	8	22.86	62	52.99	70	46.05	**0.002** ^a^
Breslow thickness (mm)	(n = 27)		(n = 55)		(n = 82)		
Median (25–75th percentile)	2.1	(1.1–5.1)	0.85	(0.44–3)	1.14	(0.6–3.3)	**0.002** ^b^
≤1 mm	5	18.52	31	56.36	36	43.9	**0.001** ^a^
>1–2 mm	8	29.63	6	10.91	14	17.07	**0.034** ^a^
>2–4 mm	6	22.22	10	18.18	16	19.51	0.664 ^a^
>4 mm	8	29.63	8	14.55	16	19.51	0.105 ^a^
Ulceration	8	22.86	12	10.26	20	13.16	0.053 ^a^
Lymphocyte infiltration							
Peritumoral	19	54.29	41	35.04	60	39.47	**0.041** ^a^
Intratumoral	13	37.14	24	20.51	37	24.34	**0.044** ^a^
Tumor mitotic rate ^+^	(n = 27)		(n = 55)		(n = 82)		
Median (25–75th percentile)	3	(1–7)	1	(0–4)	1	(0–5)	**0.012** ^a^
<1	4	14.81	25	45.45	29	35.37	**0.006** ^a^
≥1	23	85.19	30	54.55	52	64.63	
Regression	10	28.57	26	22.22	36	23.68	0.438 ^a^
Underlying histologic lesion							
Common nevus	4	11.43	12	10.26	16	10.53	0.507 ^a^
Congenital pattern melanocytic nevus	1	2.86	0	0	1	0.66	
Solar elastosis	15	42.86	52	44.44	67	44.08	0.868 ^a^

^a^: Chi-squared test; ^b^: U-Mann–Whitney; ^+^: mitosis/mm^2^; CHNM: cutaneous head and neck melanoma; LMM: lentigo maligna melanoma; SSM: superficial spreading melanoma; NM: nodular melanoma; *p*: *p*-value; and n: number of populations in the sample. Significant *p*-values in bold.

**Table 2 jcm-12-07643-t002:** Univariable and multivariable analysis of prognostic factors for recurrence and melanoma-specific death in cutaneous head and neck melanoma.

	Recurrence								Melanoma-Specific Death							
	Univariable Analysis				Multivariable Analysis				Univariable Analysis				Multivariable Analysis			
	HR	CI		*p*	HR	CI		*p*	HR	CI		*p*	HR	CI		*p*
Scalp	9.87	2.6	37.27	**0.001**	12.8	2.51	65.94	**0.002**	5.94	0.99	35.77	0.052				
Woman	1.29	0.39	4.28	0.667					0.93	0.15	5.59	0.935				
Age	1.04	0.98	1.09	0.201					1.08	0.97	1.19	0.15				
Histologic subtype	1.96	1.18	3.27	**0.01**	1.01	0.38	2.711	0.98	2.5	1.19	5.23	**0.015**	1.07	0.33	3.55	0.908
Breslow index	1.12	0.99	1.28	0.076					1.09	0.86	1.38	0.472				
Ulceration	6.11	1.86	20.06	**0.003**	2.79	0.62	12.53	0.18	6.04	0.99	36.55	0.05				
Lymphocyte infiltration																
Peritumoral	4.43	1.17	16.69	**0.028**	6.02	0.98	36.85	0.052	5.63	0.63	50.53	0.129				
Intratumoral	0.89	0.19	4.15	0.887					2.71	0.45	16.41	0.279				
Tumor mitotic rate	1.22	1.06	1.39	**0.005**	1.18	0.97	1.42	0.093	1.52	1.14	2.03	**0.005**	1.51	1.09	2.08	**0.013**
Regression	0.76	0.16	3.53	0.73					2.25	0.38	13.47	0.374				
Elastosis	1.01	0.31	3.32	0.99					4.65	0.52	41.63	0.169				
Sentinel lymph node	2.79	1.09	7.12	**0.032**					1.78	0.53	5.98	0.352				

HR: hazard ratio; CI: confidence interval; and *p*: *p*-value. Significant *p*-values in bold.

## Data Availability

The data presented in this study are available on request from the corresponding author.

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
