# Peer review of "Scalp Melanoma: A High-Risk Subset of Cutaneous Head and Neck Melanomas with Distinctive Clinicopathological Features"

_jcm, 2023, doi:10.3390/jcm12247643_

Round 1

Reviewer 1 Report

Comments and Suggestions for Authors

The manuscript presents the results of a two institute retrospective study that aimed to investigate the impact of location of H&N melanomas (scalp vs non-scalp) on recurrence and survival. A total of 152 patients were included. I have some comments below I believe would improve the paper.

Abstract:
- Please state the number of the full cohort in the abstract (152)

Introduction:
- “BANS”: I would not encourage the introduction of abbreviations that are not widely used in literature, and only twice in the paper.
- There is only 1 referral paper: multiple studies have shown that patients with melanomas located on the arms have a favourable prognosis
- “…however they concentrate between 11 to 26.7% of all cutaneous melanomas”; is this statement based on large, epidemiologic studies? If not, I would be careful with stating these percentages

Results:
- “35 (23%) developed SM”; the use of the word developed is a bit odd here. It sounds like SM develops from CHNM.
- Table 1: I would put the whole cohort (N=152) in the first row, and mention it is the whole cohort. Draw a thick line then, and then have the split cohort. In that way, it is also clearer that the p-value is between n=35 and n=117
- I hypothesize that LMM is a huge confounder in this dataset; it would be interesting to see the median Breslow thickness for each subtype in SM vs non-scalp CHNM
- “In univariate analysis…”; should be univariable (same as heading of Table 2)
- Kaplan-Meier curves are informative and look nice, however you always have to be careful not to translate this univariable analysis into a causal one. (Fig 1 + 2)
- Regarding the Cox model: how are the 5 variables that are taken along in the analysis selected? It seems based on the p-value of the univariable analysis, but this is not the correct way. (It leads to a selection bias on your own data). The correct method would be to base the selection on literature, and on “logical”/theoretical thinking about which variables can play a role in the interaction between the determinant and the selected outcome.    
- Only 11 patients experience a recurrence. There are 5 variables now used in the Cox model. To ensure a meaningful and statistically relevant outcome, a ratio of 1:10 (= 1 variable in the model for every 10 events) is preferred. In other words: the dataset is too small for meaningful multivariable analyses.
- What is the number of patients experiencing death? Same comment as above.

Conclusion:
- Statements in conclusion need to be revised according to the changes indicated above. Please be very careful with claiming causality between SM and certain variables based on data like this. I would rather encourage the authors to keep it descriptive, given the small number of patients that were included and the very low number of events.

Comments on the Quality of English Language

Please check English spelling throughout the paper – e.g. melanomas arising in the scalp à on the scalp, melanomas located in BANS region à located in the BANS region

Author Response

Thank you for your review. We have considered all your suggestions. Please see the attachment

Reviewer 2 Report

Comments and Suggestions for Authors

Dear Authors,

I have reviewed your manuscript titled "Scalp melanoma: a high-risk subset of cutaneous head and neck melanomas with distinctive clinicopathological features." This paper investigates the clinical characteristics of scalp melanoma, compares it to non-scalp head and neck melanomas, and aims to identify prognostic features while underscoring the importance of evaluating this area in high-risk patients.

Overall, your manuscript is well-written, well-structured, and employs an appropriate study design. The references are well-constructed and up-to-date. However, there are several points that could be addressed to strengthen the paper:

1. Abstract:

  • Review the section on the study population for clarity and completeness.
  • Consider making this section more succinct (particularly lines 17-22).

2. Keywords:

  • Ensure that the keywords enhance the paper's visibility and do not duplicate those in the title. Using MeSH terms is recommended.
  • Replace the first two keywords with general terms such as "skin cancer," "melanoma," and "prognosis."

3. Materials and Methods:

  • Provide a clear description of the data collection process, specifically what data (Table 1) you collected, including epidemiological and histological features.
  • Clarify whether indeterminate malignant tumors (SAMPUS, MELTUMP, AST) were included in the database or not.
  • Specify whether all included cases were de novo melanomas or if there were nevus-associated melanomas.
  • Explain the rationale behind choosing median values and not mean values for age, Breslow, mitotic rate, and follow-up period.
  • Address the independence of subsets in cases where scalp melanoma (SM) and non-scalp head and neck melanoma (CHNM) may overlap in one patient.
  • Justify your choice of the Chi-square test over the Fisher exact test for categorical data. The further provides just estimated results, were the latter exact ones.

4. Results - Table 1:

  • Make the data clearer by specifying M/F distribution instead of just "Woman."
  • Clarify the meaning of "non-melanoma skin cancer" and whether it refers to a personal history of skin cancer or a concurrent cancer diagnosed during the follow-up period.
  • Consider using "nevi count" instead of "common nevi."
  • Explain the significance level you consider for "p" values.
  • Ensure consistency in capitalization (e.g., "in situ").
  • Describe the criteria used to differentiate between common nevi and congenital nevi.
  • Clarify which histologic subtype was associated with a higher risk for recurrence, and ensure that "higher mitotic rate" is what you intended to convey (instead of "mitotic rate").
  • Provide legends for tables introducing abbreviations used (e.g., n, p, HR, CI).
  • Consider highlighting significant p-values in tables.
  • Explain the reference to "Table II" (line 131). Shouldn't it be Table 2?
  • Introduce the abbreviation "HR" in the text.
  • Address whether you investigated the relationship mentioned in lines 145-147.
  • Explain the reason for the absence of desmoplastic subtypes in your database.
  • Elaborate on why SM recurred more frequently.
  • Clarify why the recurrence rate in SM was higher than in other locations (lines 175-177).
  • Discuss the observed high recurrence rate in lentiginous melanoma (LM/LMM) compared to your results (as per Collgros et al., https://onlinelibrary.wiley.com/doi/10.1111/jdv.17135).
  • Reference the current AJCC guidelines (lines 191-193). Breslow needs capitalization.
  • Explain the differences in SLNB between your results and previous reports (lines 194-195).

5. Conclusions:

  • Restructure this section to avoid duplicating the results.

6. Data Availability Statement:

  • Please consider enhancing the credibility of your study by providing a link to an external repository for your data.

7. References:

  • Address the formatting problem in the references starting from line 280.
Comments on the Quality of English Language

The paper would benefit from moderate English editing.

Author Response

Thank you for your review. We have considered all your suggestions. Please see the attachment.

Round 2

Reviewer 2 Report

Comments and Suggestions for Authors

Dear Authors,

Thank you for adressing all the issues raised. Although there are still minot typos, these can be dealt with by technical editor. I believe the paper is ready to be considered for publication. Congratulations!